# Improvements in airflow characteristics and effect on the NOSE score after septoturbinoplasty: A computational fluid dynamics analysis

**Yang Na[1], Youn-Ji Kim[1], Hyo Yeol Kim[2], Yong Gi Jung[2]\***

**1** Department of Mechanical Engineering, Konkuk University, Gwangjin-gu, Seoul, Korea, **2** Department of Otorhinolaryngology-Head and Neck Surgery, Samsung Medical Center, Sungkyunkwan University, School of Medicine, Seoul, Korea

\* ent.jyg@gmail.com

## Abstract

Septoturbinoplasty is a surgical procedure that can improve nasal congestion symptoms in patients with nasal septal deviation and inferior turbinate hypertrophy. However, it is unclear which physical domains of nasal airflow after septoturbinoplasty are related to symptomatic improvement. This work employs computational fluid dynamics modeling to identify the physical variables and domains associated with symptomatic improvement. Sixteen numerical models were generated using eight patients' pre- and postoperative computed tomography scans. Changes in unilateral nasal resistance, surface heat flux, relative humidity, and air temperature and their correlations with improvement in the Nasal Obstruction Symptom Evaluation (NOSE) score were analyzed. The NOSE score significantly improved after septoturbinoplasty, from $14.4 \pm 3.6$ to $4.0 \pm 4.2$ ($p < 0.001$). The surgery not only increased the airflow partition on the more obstructed side (MOS) from $31.6 \pm 9.6$ to $41.9 \pm 4.7\%$ ($p = 0.043$), but also reduced the unilateral nasal resistance in the MOS from $0.200 \pm 0.095$ to $0.066 \pm 0.055$ Pa/(mL·s) ($p = 0.004$). Improvement in the NOSE score correlated significantly with the reduction in unilateral nasal resistance in the preoperative MOS ($r = 0.81$). Also, improvement in the NOSE score correlated better with the increase in surface heat flux in the preoperative MOS region from the nasal valve to the choanae ($r = 0.87$) than in the vestibule area ($r = 0.63$). Therefore, unilateral nasal resistance and mucous cooling in the preoperative MOS can explain the perceived improvement in symptoms after septoturbinoplasty. Moreover, the physical domain between the nasal valve and the choanae might be more relevant to patient-reported patency than the vestibule area.

## Introduction

Nasal septal deviation (NSD) is an asymmetry in the shape of the nasal septum, and anatomical closeness to the inferior turbinate can cause airflow modifications and nasal airway

**Data Availability Statement:** https://datadryad.org/stash/dataset/doi:10.5061/dryad.nk98sf7wj

**Funding:** NY received academic funding for this research. This work was supported by the National

Research Foundation of Korea (NRF) grant funded
by the Korean Government (MSIT, No. NRF-
2020R1A2C1005128).

**Competing interests:** The authors have declared
that no competing interests exist.

obstruction (NAO) [1, 2]. Septoplasty is a procedure that corrects an asymmetric nasal septum
to the midline, and turbinoplasty is a procedure that reduces the volume of the inferior turbi-
nate. In almost all NSD patients, septoplasty and turbinoplasty are performed together; there-
fore, the term septoturbinoplasty is widely used. Predicting how various types of NSD will
disturb airflow in the nasal cavity is complex, and understanding the driving factors that pro-
duce symptoms such as NAO is limited. However, despite the complexity, identifying the links
between objective measures and patient sensations of patency is essential to understanding the
cause of patient symptoms and to establishing an accurate surgical plan.

The narrow side of the nasal airway is expected to benefit from surgical correction of the
deviated septum in terms of airflow regulation by reducing the cross-sectional imbalance of
bilateral nasal airways. Furthermore, because nasal physiology is also influenced by the air con-
ditioning characteristics of the nasal cavity [3–8], the optimized outcome of septoturbinoplasty
can be evaluated in terms of heat capacity modification as well as reallocation of flow partition
between the airways.

Several studies using rhinomanometry and acoustic rhinometry have tried to relate subjec-
tive patency to nasal resistance (NR) without consistent success [9–11], and multiple studies
have focused on how changes in mucosal temperature affect the nasal sensation of airflow [12–
14]. The computational fluid dynamics (CFD) technique has become a valuable tool in charac-
terizing the airflow and air conditioning capacity of the nasal cavity because it overcomes the
difficulty of measuring the pressure and intranasal epithelial surface temperature *in vivo*.
Although CFD has been used to investigate NAO resulting from NSD [15–18], few studies
have used it to predict flow modification after septoturbinoplasty [19–22]. Therefore, changes
in airflow characteristics after surgical correction of NSD are not well understood.

Sullivan et al. [22] suggested that the patient's perception of better nasal patency after NAO
surgery correlated well with increased heat loss in the vestibule area of the preoperative more
obstructed side (MOS). However, because the structural change is not limited to the vestibule
area after surgery, gains in mucosal cooling from the domain of the nasal cavity beyond the
nasal vestibule need to be more clearly assessed. In addition, although the most significant
structural change after septoturbinoplasty is posterior to the vestibular area, the latent heat
resulting from water vapor transfer from the epithelial surface of that area has not yet been
elucidated.

Therefore, it is necessary to investigate the changes in airflow characteristics that occur
after septoturbinoplasty throughout the entire nasal cavity, from the anterior septum to the
choanae.

Our objective in this study is to use CFD to investigate two critical principles of improve-
ments in subjective patency after septoturbinoplasty. The first is how improvements in
patient-reported patency after septoturbinoplasty correlate with changes in physiological vari-
ables such as flow partition, NR, and air conditioning capacity. The second is determining
which anatomical domains of airflow (e.g., vestibule or nasal cavity posterior to the vestibule)
are most relevant in improving patient-reported outcomes.

## Patients and methods

### Patient recruitment and surgical techniques

The Institutional Review Board at Samsung Medical Center approved this study (SMC 2021-
05-120-001). Subjects' data were obtained retrospectively, and informed consent was waived
after IRB approval. All data was processed following the institution's anonymization regula-
tions. Between January 2019 and April 2020, adult patients aged 18 years or older who under-
went septoturbinoplasty by a single experienced surgeon (HY Kim) at Samsung Medical

Center were screened for this study. Data on thin-section computed tomography (CT) scan images (0.625 mm thickness) were obtained preoperatively and three months postoperatively. The subjects completed the Nasal Obstruction Symptom Evaluation (NOSE) scale questionnaire on the same days of the CT scans. Septoturbinoplasty was performed on subjects diagnosed with symptomatic NSD who did not respond to appropriate medical treatment, including intranasal topical steroids and antihistamines, or who showed insufficient symptom control. Both septoplasty and turbinoplasty were recommended to secure the patient's nasal airway as much as possible, and septoturbinoplasty was performed in all subjects after obtaining consent. Subjects who received additional procedures, such as endoscopic sinus surgery or revision septoturbinoplasty, were excluded from this study. Thus, 17 subjects who met those selection criteria were chosen.

Before constructing a 3-dimensional (3D) model for CFD analysis, three independent otolaryngologists who did not participate in this study reviewed the pre- and postoperative CT scans and evaluated the mucosal thickness and degree of swelling to examine the presence of the nasal cycle. The seven subjects who exhibited a significant difference in mucosal swelling were excluded because it was judged that their mucosal status was affected by the nasal cycle. Therefore, the 3D numerical domains were constructed using pre- and postoperative CT images obtained from 10 subjects. Two further subjects were excluded due to difficulty in segmentation because of a large amount of secretion in the CT images. Thus, 16 CFD models were obtained from eight subjects (six males and two females) and analyzed. The average age of the subjects was 35.5 years, with a range of 19–72 years.

All patients were treated with the same surgical techniques during their operations. First, a modified Killian incision was performed on the concave side of the curvature to approach the nasal septum. After separating the quadrangular cartilage from the perpendicular ethmoid plate, vomer, and maxillary crest, a keystone mattress suture was performed to correct the junction between the deviated perpendicular plate and the septal cartilage. Next, the deviated septal cartilage was corrected with a barbed suture using STRATAFIX$^{TM}$ (Ethicon, Somerville, NJ). Bilateral turbinoplasty was performed using a microdebrider through the submucosal pocket to reduce the volume of the inferior turbinate, and the inferior turbinate was pushed laterally with the in and out fracture technique.

## Nasal cavity modeling

Numerical models for the nasal cavity were produced using CT scan data from eight subjects (S1–S8) who underwent septoturbinoplasty to correct symptomatic NSD. The segmentation procedure for the subjects' pre- and postoperative CT scan images was conducted using Mimics v22.0 software (Materialise, Leuven, Belgium). The output file in a stereolithography format was further processed using Geomagic Wrap 2021 (3D Systems, Rock Hill, SC) to produce a 3D computational volume model. The paranasal sinuses were removed during segmentation because we assumed that their presence did not significantly affect nasal airflow [4, 6, 20–23]. The pre- and postoperative CT scan images of subject S1 are presented as examples in Fig 1.

As shown in Fig 2, the nasal cavity models included 3D-rendered models of the face around the nose and nostrils to induce a more realistic inflow through the nostrils. Also, a semi-spherical chamber was placed in front of the face to describe the atmospheric inlet pressure boundary condition conveniently. Finally, we divided the entire nasal airway from the nostril to the nasopharynx into three parts so that we could separately analyze the effect of each anatomical domain on airflow characteristics. Here, Area-1, Area-2, and Area-3 represent the vestibule area, the nasal cavity area (including the inferior, middle, and superior turbinates, with the choanae as the posterior boundary), and the nasopharynx area, respectively.

**A**   (coronal view)                    (axial view)

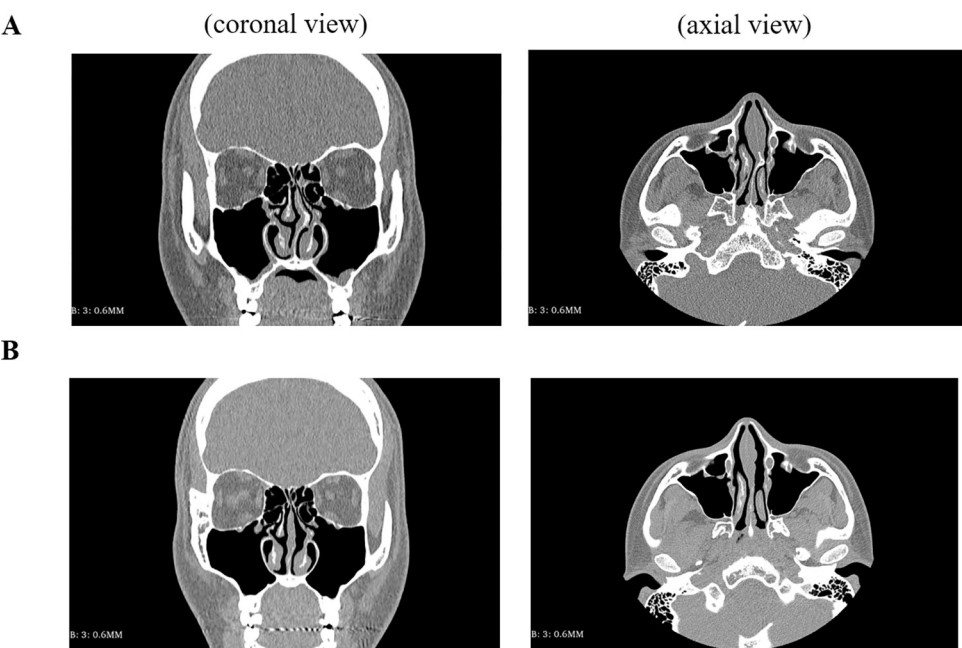

**B**

**Fig 1. Pre- and postoperative CT scan images for subject S1.** (A) Preoperative CT image. (B) Postoperative CT image.

## Numerical methodology

A set of governing equations consisting of the continuity, momentum, energy, and species equations for a mixture of air and water vapor in the airway was solved using ANSYS/Fluent 2021R1 (Canonsburg, PA) and the wall model [23, 24]. Although the literature shows some variations [25–27], an average temperature of 25˚C and 35% relative humidity (RH) were assumed as the ambient condition.

Fluent Meshing 2021R1 was used to generate a combined mesh system with seven prism layers along the epithelial surface, with a growth rate of 1.1 and polyhedral elements in the airway space. With reference to the work of Siu et al. [28], approximately 4.2 million mesh elements were used for all the cavity models in this study. In addition, spatial discretization of the governing equations was performed with a second-order central differencing scheme to reduce the dissipation error.

The pre- and postoperative models can be directly compared in several ways. For example, the same trans-nasal pressure decreases, or the same target airflow rates could be imposed. It was assumed that maintaining the same breathing effort through the nasal cavity, which is related to the trans-nasal pressure drop, was more relevant to the objective of this study than was maintaining the same target airflow rate. Therefore, the same trans-nasal pressure decrease between the inlet and the nasopharynx was imposed for both pre- and postoperative models of each subject, as proposed by Sullivan et al. [22], to compare models before and after surgery. In this procedure, the outlet pressure, which was adjusted to yield a target bilateral airflow rate of 250 mL/s in the postoperative model, was also imposed for computation with the preoperative model. Many previous numerical investigations suggest that turbulent flow features are not noticeable at the flow rates considered in this study [27, 29, 30]; thus, a laminar flow regime was assumed.

Two representative planes were chosen to evaluate quantities such as pressure, air temperature, and RH, as presented in Fig 3. Plane-1 and Plane-2 were located immediately before and

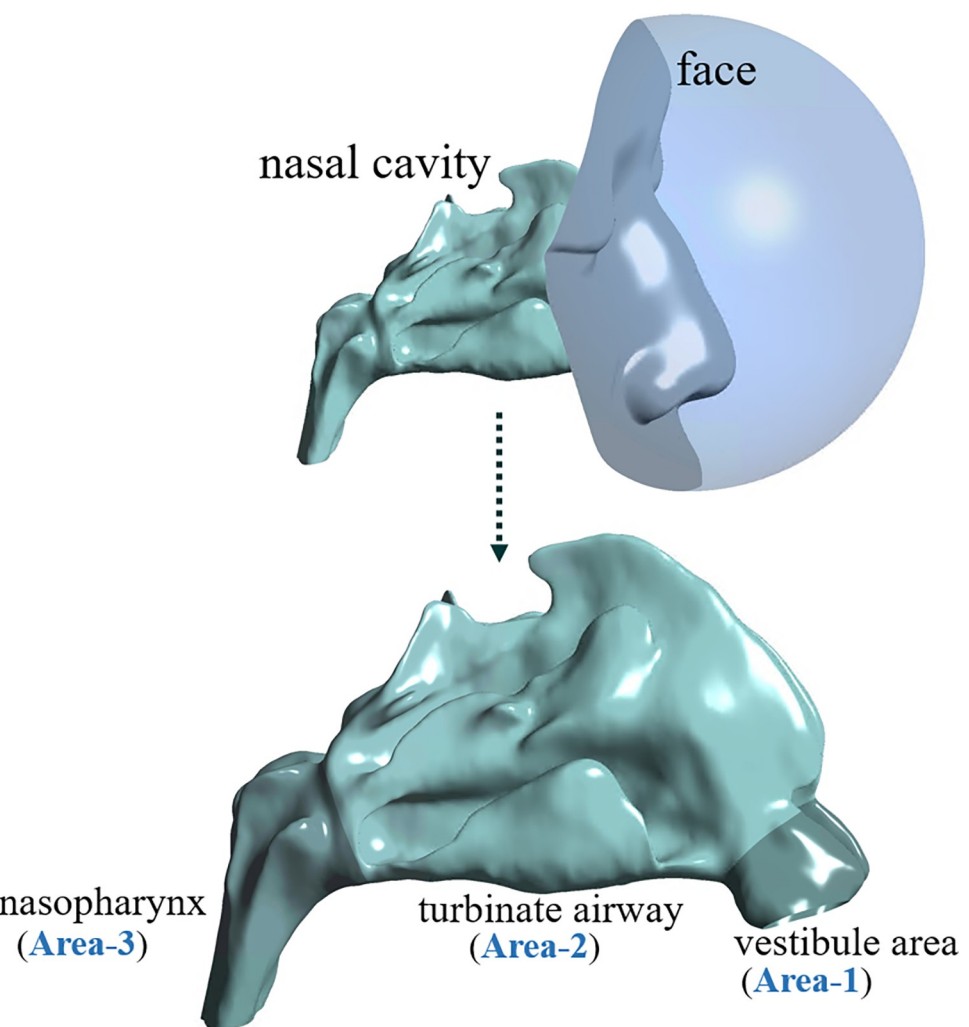

**Fig 2. Configuration of the numerical model of the nasal cavity for subject S1.**

after the choanae, respectively, and the mass-weighted average of physical variables was calculated in those planes.

Two-tailed paired Student t-tests (Excel v.2110, Redmond, WA) were used to assess the statistical difference between the pre- and postoperative values. Differences were estimated to be statistically significant when the p-value was <0.05. The Pearson's correlation coefficient (Pearson $r$-value) was used to assess the statistical correlations (Origin Pro 2016, Northampton, MA) between improvements in the NOSE score and changes in the CFD variables.

## Results

### Patient-reported symptoms

The NOSE score results evaluating patient-reported symptoms before and after surgery are summarized in Table 1. The NOSE score improved significantly after septoturbinoplasty, from 14.4±3.6 to 4.0±4.2 (p < 0.001). When sub-analyzing each subject, subjects S1 to S3 improved by 15.7±2.1 on the NOSE scale, which was significantly higher than the improvement of 7.2

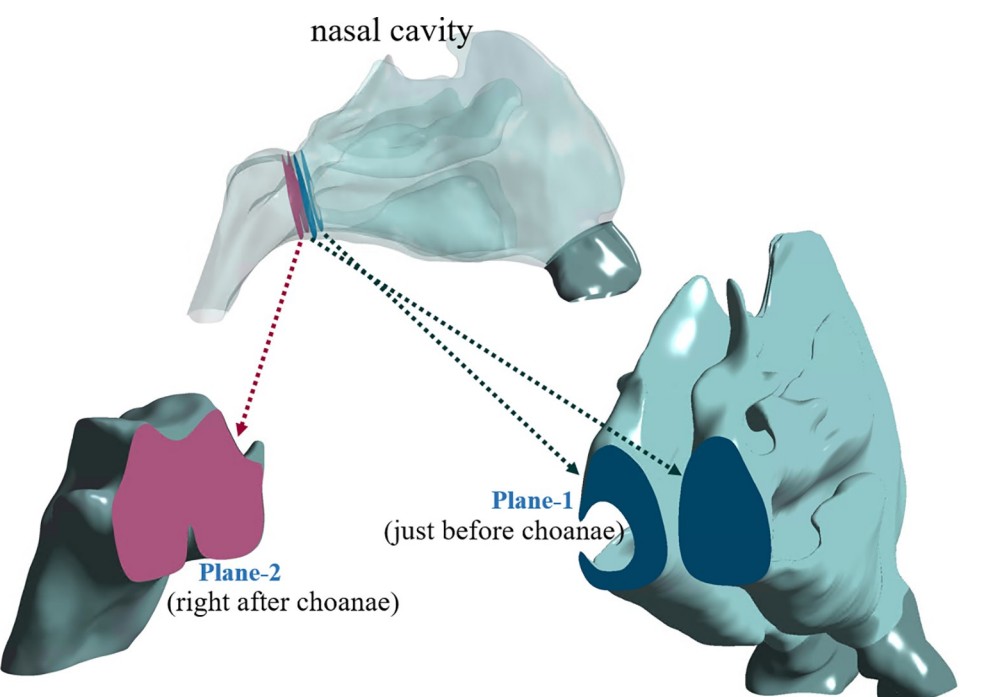

**Fig 3. Two representative planes for evaluating physical variables around the choanae.**

±2.3 in subjects S4 to S8 (p = 0.004). Therefore, subjects S1 to S3 were named the good response group.

## Flow partition ratio and nasal resistance

The airflow partition in the MOS improved significantly after surgery, from 31.6±9.6 to 41.9 ±4.7% (p = 0.043) (Tables 2 and 3). The flow partition ratio between the less obstructed side (LOS) and the MOS was high (3.65–4.05) for subjects S1–S3 before surgery, and it was significantly reduced after surgery (1.31–1.45). Note that these three subjects (S1–S3) showed the most significant improvement in NOSE score (Table 1).

After surgery, the unilateral NR in the MOS decreased from 0.200±0.095 to 0.066±0.055 (p = 0.004). NR was also reduced bilaterally after the surgery, as shown by the mean of 0.027

**Table 1. Summary of the NOSE score before and after septoturbinoplasty.**

| Subject | NOSE score | | Difference in NOSE score |
|---|---|---|---|
| | **Preoperative** | **3 months postoperative** | |
| S1 | 18 | 0 | 18 |
| S2 | 15 | 0 | 15 |
| S3 | 17 | 3 | 14 |
| S4 | 10 | 0 | 10 |
| S5 | 20 | 12 | 8 |
| S6 | 13 | 5 | 8 |
| S7 | 9 | 3 | 6 |
| S8 | 13 | 9 | 4 |
| **Mean ± SD** | 14.4 ± 3.9 | 4.0 ± 4.5 | 10.4 ± 4.8 |
| **p-value; pre- & post-op** | < 0.001 | | |

**Table 2. Variations in the flow partition in the MOS and the NR before and after septoturbinoplasty.** (NR, MOS, and LOS denote nasal resistance, more obstructed and less obstructed sides, respectively).

| Subject | | Flow partition in the MOS at nostrils (%), (ratio of LOS to MOS) | Unilateral NR in the MOS between nostril and choanae (Pa/(mL·s)), (ratio of MOS to LOS) | Bilateral NR between inlet and choanae (Pa/(mL·s)) |
|---|---|---|---|---|
| S1 | pre-op | 20.2 (3.95) | 0.312 (5.24) | 0.0680 |
| | post-op | 40.8 (1.45) | 0.055 (1.85) | 0.0321 |
| S2 | pre-op | 21.5 (3.65) | 0.299 (4.36) | 0.0663 |
| | post-op | 43.2 (1.31) | 0.061 (1.59) | 0.0326 |
| S3 | pre-op | 19.8 (4.05) | 0.299 (6.82) | 0.0616 |
| | post-op | 41.2 (1.43) | 0.091 (1.91) | 0.0459 |
| S4 | pre-op | 44.1 (1.27) | 0.074 (1.32) | 0.0350 |
| | post-op | 44.6 (1.24) | 0.010 (1.71) | 0.0098 |
| S5 | pre-op | 39.8 (1.51) | 0.211 (1.90) | 0.0973 |
| | post-op | 30.2 (2.31) | 0.189 (5.04) | 0.0661 |
| S6 | pre-op | 43.1 (1.32) | 0.151 (1.36) | 0.0681 |
| | post-op | 43.5 (1.30) | 0.029 (1.66) | 0.0188 |
| S7 | pre-op | 34.4 (1.91) | 0.053 (3.69) | 0.0237 |
| | post-op | 46.9 (1.13) | 0.007 (1.42) | 0.0156 |
| S8 | pre-op | 29.8 (2.36) | 0.204 (4.95) | 0.0689 |
| | post-op | 44.6 (1.24) | 0.087 (1.55) | 0.0512 |
| **Mean ± SD** | pre-op | 31.6 ± 10.3 | 0.200 ± 0.101 | 0.061 ± 0.023 |
| | post-op | 41.9 ± 5.1 | 0.066 ± 0.059 | 0.034 ± 0.019 |

**Table 3. Variables related to flow partition and nasal resistance and the Student's t-test results between the pre- and postoperative models (denoted by p-value).** (NR, MOS, and LOS denote nasal resistance, more obstructed and less obstructed sides, respectively).

| Variables | Preoperative | Postoperative | p-value |
|---|---|---|---|
| • flow partition (%)in MOS | 31.6 ± 10.3 | 41.9 ± 5.1 | 0.043 |
| • flow partition ratio, LOS/MOS | 2.50 ± 1.20 | 1.43 ± 0.37 | 0.052 |
| • unilateral NR (Pa/(mL·s)) in MOS | 0.200 ± 0.101 | 0.066 ± 0.059 | 0.004 |
| • unilateral NR (Pa/(mL·s)) in LOS | 0.063 ± 0.034 | 0.030 ± 0.018 | 0.044 |
| • bilateral NR (Pa/(mL·s)) | 0.061 ± 0.023 | 0.034 ± 0.019 | <0.001 |
| • unilateral NR ratio, LOS/MOS | 3.71 ± 2.02 | 2.09 ± 1.20 | 0.129 |

**Table 4. Correlations between the NOSE score and improvements in the NOSE score and variables related to the flow partition and nasal resistance (denoted by Pearson *r*-value).** Data exhibiting the statistical significance of correlation (p < 0.05) are shaded. (NR, MOS, and LOS denote nasal resistance, more obstructed and less obstructed sides, respectively).

| Correlation variable 1 | Correlation variable 2 | Pearson *r*-value |
|---|---|---|
| NOSE score | • flow partition, MOS | −0.67 |
| | • flow partition ratio, LOS/MOS | 0.65 |
| | • unilateral NR, MOS | 0.86 |
| | • unilateral NR, LOS | 0.66 |
| | • bilateral NR | 0.84 |
| | • unilateral NR ratio, LOS/MOS | 0.60 |
| Improvement in NOSE score | • reduction in unilateral NR (preoperative MOS) | 0.81 |
| | • reduction in bilateral NR | 0.31 |

±0.012 Pa/(mL·s) (p < 0.001). Among the patients, subjects S1–S3 (the good response group) not only showed the highest unilateral NR ratios (4.36–6.82), but also exhibited a considerably large unilateral NR in the preoperative MOS (0.299–0.312 Pa/(mL·s)).

The correlations between the NOSE score (as well as the improvements in NOSE score) and several variables are summarized in Table 4, with the shaded data indicating statistically significant correlations (p < 0.05). Both the unilateral NR in the MOS (*r* = 0.86) and the bilateral NR (*r* = 0.84) showed a high correlation with the NOSE score. However, the flow partition ratio and unilateral NR ratio did not exhibit a statistically significant difference from before to after the surgery (p = 0.052 and 0.129, respectively, as shown in Table 3), although they exhibited moderate correlations with the NOSE score (*r* = 0.65 and 0.60, respectively). Because the correlations between the NOSE score and both the unilateral and bilateral NRs were similar, the improvements in NOSE score and NR reduction were further explored, as shown in Fig 4. Although the reduction in bilateral NR did not show any meaningful correlation with improvement in NOSE score (*r* = 0.31), reduction in the unilateral NR in the MOS (defined preoperatively) exhibited a strong correlation (*r* = 0.81).

## Air conditioning capacity

The mass-weighted average RH and inhaled air temperature in two representative planes (defined in Fig 3) were evaluated before and after surgery (Table 5). In all subjects, the RH immediately after the choanae (in Plane-2, defined in Fig 3) decreased after surgery from 94.9 ±3.3 to 86.2 ± 4.5% (p = 0.008). The data in Table 5 suggest that the reduction in RH after surgery was due primarily to insufficient humidification in the LOS of the postoperative models. Reflecting the similarity between the transport phenomena of heat and water vapor, the behavior of the air temperature was quite similar to that of RH in that it decreased after the choanae (in Plane-2), from 32.3±0.7 to 31.0±0.5˚C (p = 0.013).

In the presence of a large amount of water vapor transfer from the epithelial surface to the inhaled air, the contribution of latent heat can be significant in the nasal cavity (i.e., Area-2, defined in Fig 2) except in the vestibule area (i.e., Area-1), where locally distributed squamous cells along the surface allow almost no water transport to the air. Table 6 shows the variables related to the surface heat flux (both sensible and latent) and their statistical differences before and after surgery in separate domains of Area-1 and Area-2. Regardless of inclusion of the latent heat in Area-2, no statistical difference between the pre- and postoperative models was noted in the surface heat flux in the LOS (p > 0.05).

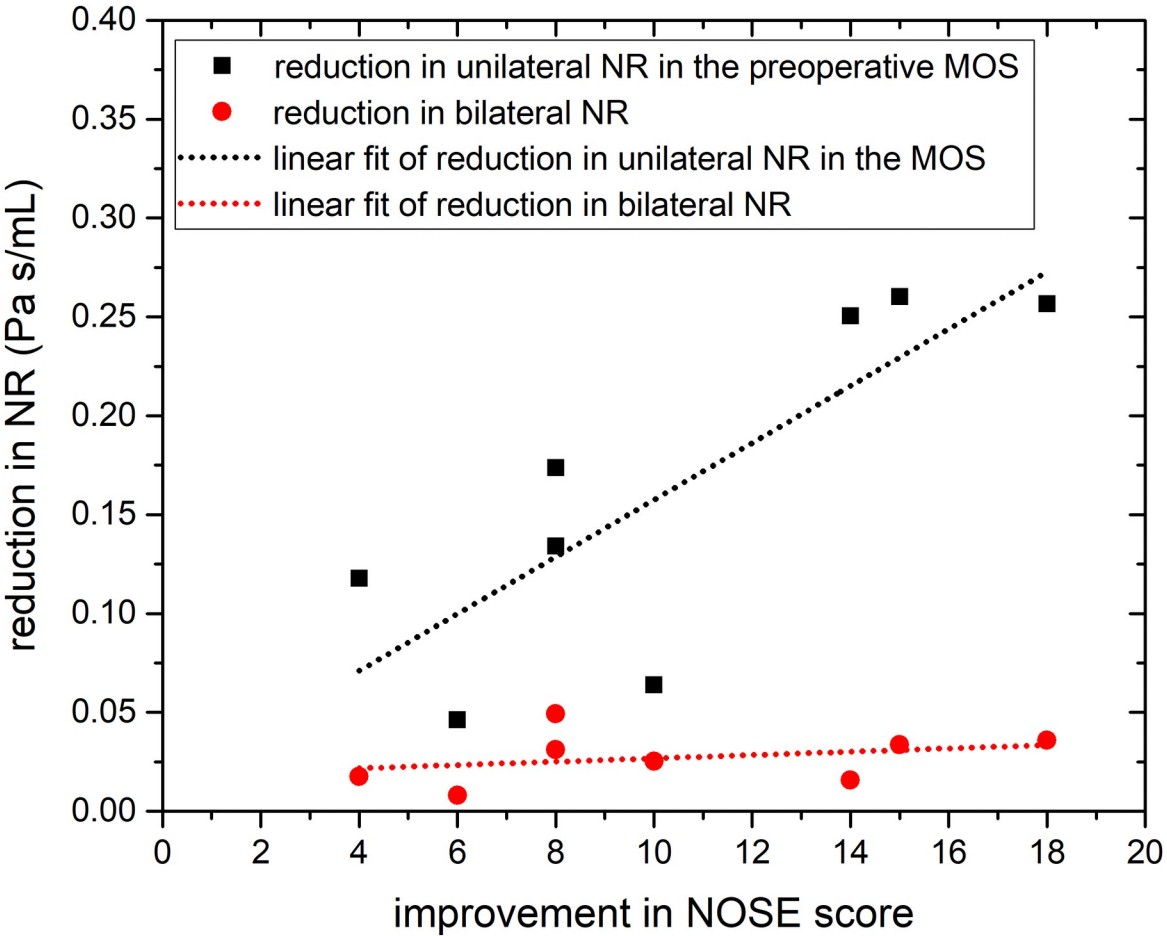

**Fig 4. Correlation between improvement in the NOSE score and reduction in nasal resistance (NR).**

The correlation between improvement in NOSE score and the increase in surface heat flux is summarized in Table 7, with the shaded numbers indicating the variables with a statistically significant correlation. Whereas the improvement in the NOSE score correlated moderately with an increase in sensible surface heat flux in the preoperative MOS in Area-1 ($r = 0.63$), it correlated highly with the increase in surface heat flux in Area-2 ($r = 0.87$–0.88). Note, however, that statistical significance of correlation was not found in the Area-1. The high correlation and statistical significance in Area-2 persisted regardless of inclusion of latent heat (also shown in Fig 5).

**Table 5. Comparison of relative humidity and air temperature before and after septoturbinoplasty.** (RH, MOS, and LOS denote relative humidity, more obstructed and less obstructed sides, respectively).

| Subject | | RH (%) | | RH (%) Plane-2 | Air temperature (°C) | | Air temperature (°C) Plane-2 |
|---|---|---|---|---|---|---|---|
| | | Plane-1, MOS | Plane-1, LOS | | Plane-1, MOS | Plane-1, LOS | |
| Mean | pre-op | 98.6 ±2.5 | 93.4 ±5.0 | 94.9 ± 3.5 | 33.1 ± 0.5 | 31.9 ±1.0 | 32.3 ± 0.7 |
| | post-op | 89.3 ±7.8 | 83.5 ±5.0 | 86.2 ± 4.9 | 31.5 ± 0.9 | 30.7 ±0.4 | 31.0 ± 0.5 |
| p-value | pre- & post-op | 0.024 | 0.003 | 0.008 | 0.010 | 0.016 | 0.013 |

**Table 6. Surface heat flux in the two physical domains defined in Fig 2.** Data exhibiting a statistical difference between the pre- and postoperative models (p < 0.05) are shaded. (MOS and LOS denote more obstructed and less obstructed sides, respectively).

| Surface heat flux (W/m$^2$) | Physical domain | | | | | |
|---|---|---|---|---|---|---|
| | Area-1 | | Area-2 | | Area-1 + Area-2 | |
| | Pre-op | Post-op | Pre-op | Post-op | Pre-op | Post-op |
| • sensible, MOS | 125.9 ± 39.3 | 174.4 ± 16.5 | 43.0 ± 16.8 | 68.8 ± 15.3 | 47.7 ± 18.4 | 75.1 ± 14.9 |
| • sensible, LOS | 178.5 ± 66.5 | 196.5 ± 45.0 | 86.2 ± 27.8 | 83.6 ± 18.1 | 91.8 ± 29.5 | 90.3 ± 18.0 |
| • sensible, MOS + LOS | 152.3 ± 45.7 | 185.4 ± 25.7 | 65.2 ± 19.8 | 76.4 ± 13.5 | 70.5 ± 21.1 | 82.9 ± 14.1 |
| • sensible + latentMOS | - | - | 528.4 ± 183.9 | 803.4 ± 174.6 | 503.4 ± 175.3 | 765.2 ± 161.6 |
| • sensible + latentLOS | - | - | 979.9 ± 303.2 | 944.7 ± 199.6 | 931.5 ± 290.9 | 855.6 ± 251.7 |
| • sensible + latentMOS + LOS | - | - | 760.6 ± 219.9 | 876.0 ± 160.0 | 724.0 ± 224.3 | 834.7 ± 149.5 |

## Discussion

Anatomic deformity of the septum can render the airflow distribution between the left and right sides highly asymmetric, and the resulting skewed flow distributions can create any of several physiological issues, depending on the degree of flow partition. Although septoturbinoplasty is an effective surgical technique to recover the impaired physiological functions of the nasal cavity, incomplete knowledge about the relationships between patient-reported patency and objective measures makes it difficult to predict surgical outcomes.

Traditionally, NR was the primary objective measure used to assess NAO, and it was assumed to directly affect patients' subjective patency. In addition, the sensation of mucosal cooling in the nasal cavity was assumed to be an essential factor that affects patient-reported nasal obstruction. Previous studies investigated the hypothesis that subjective patency correlated better with mucosal cooling than with NR [12–14, 18]. The role of the vestibule area in sensing nasal patency was also reported by Willatt and Jones [31] and Sullivan et al. [22]. Those studies on nasal sensation prompted more quantitative investigations to refine their findings.

In this study, subjective measures of symptoms (NOSE score) and NR both decreased significantly after septoturbinoplasty. Furthermore, the high correlations of unilateral NR in the MOS and bilateral NR with the NOSE score ($r = 0.86$ and 0.84, respectively), shown in Table 4, confirmed the association between NR and the NOSE score. However, improvement in the NOSE score correlated strongly only with improvement in the unilateral NR in the preoperative MOS ($r = 0.81$), supporting the hypothesis that unilateral NR in the MOS of the preoperative model is relevant to the amount of improvement in the NOSE score. This conclusion

**Table 7. Correlation between improvements in the NOSE score and the % increase in surface heat flux (denoted by Pearson *r*-value).** Data exhibiting the statistical significance of correlation (p < 0.05) are shaded. (MOS and LOS denote more obstructed and less obstructed sides, respectively).

| % Increase of surface heat flux | Physical domain | | |
|---|---|---|---|
| | Area-1 | Area-2 | Area-1 + Area-2 |
| • sensible, MOS defined preoperatively | 0.63 | 0.87 | 0.88 |
| • sensible, LOS defined preoperatively | 0.06 | -0.06 | -0.04 |
| • sensible, MOS + LOS | 0.32 | 0.23 | 0.26 |
| • sensible + latent, MOS defined preoperatively | - | 0.88 | 0.87 |
| • sensible + latent, LOS defined preoperatively | - | -0.03 | 0.007 |
| • sensible + latent, MOS + LOS | - | 0.30 | 0.29 |

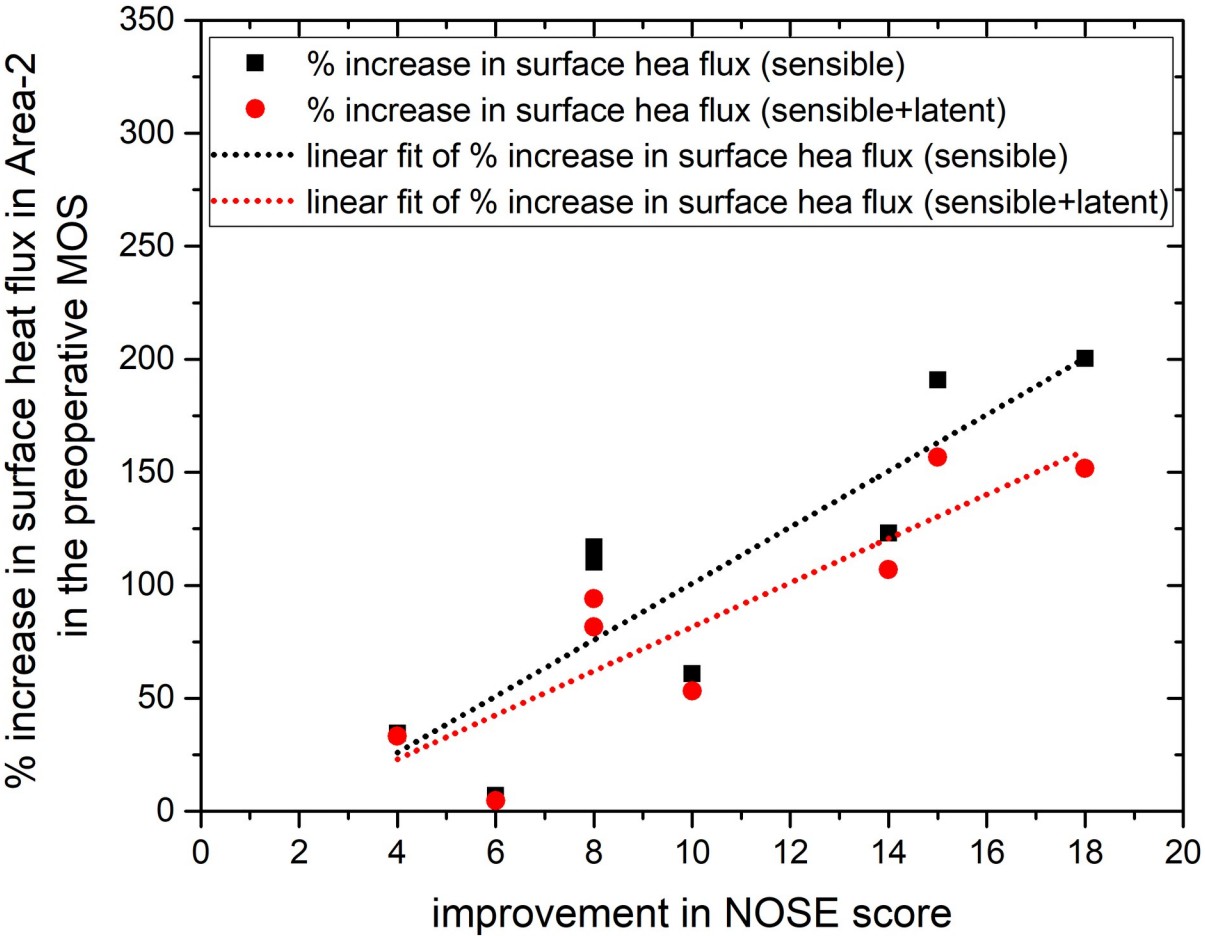

**Fig 5. Correlation between improvement in the NOSE score and the % increase in surface heat flux in Area-2 in the preoperative MOS.**

agrees with the findings of Kimbell et al. [32] and Sullivan et al. [22] that patient perception correlates with unilateral NR but not bilateral NR.

Notably, the change in the ratio of surface area to volume of the nasal cavity after surgery was much more prominent in Area-2 than in Area-1, suggesting greater structural modification in Area-2. Therefore, modification of the flow characteristics was expected to be more prominent in Area-2, which led us to hypothesize that the surface heat flux in Area-2 would be more relevant to subjective patency than that in Area-1. The significance of mucosal cooling in the vestibule area was discussed in detail by Sullivan et al. [22]. They emphasized an anatomical study that showed that the density of thermoreceptors is highest in the vestibule area (Jones et al. [12]). In this study, the surface area of Area-2 was much larger than that of Area-1 in the MOS (11.5–18.6 times larger), so the higher density of thermoreceptors in Area-1 does not necessarily imply that it has a larger number of thermoreceptors than Area-2. In fact, Table 7 shows that the percentage increase in the surface heat flux in the preoperative MOS in Area-2 correlates with the improvement in the NOSE score more strongly ($r = 0.87$–$0.88$) than that in Area-1 ($r = 0.63$). Because the improvement in the NOSE score does correlate modestly with the change in surface heat flux in Area-1, the subjective benefit is not restricted to the change in Area-2. Nevertheless, statistical significance was found only in Area-2 (Table 7), which supports our hypothesis that Area-2 is more important than Area-1.

The use of various criteria to define mucosal cooling, as discussed in several studies [17, 18, 22, 32, 33], suggests the need for more sophisticated outcome measures for surgery. Table 5 shows that RH and air temperature at the posterior end of the septum decreased for all subjects after surgery, despite the significant improvement in patient-reported patency (Table 1). Although the RH decreased after surgery, no adverse effects, such as crust formation, were observed in the postoperative clinical examinations, suggesting that the RH listed in Table 5 is tolerable. Although air temperature and RH are not appropriate objective indicators of short-term patient-reported patency, they might play a valuable role in long-term evaluations of surgical outcomes [34].

In the presence of water vapor transfer, the latent heat is frequently much larger than the sensible heat and could significantly change the physics of the airflow. Table 7, however, shows that latent heat played a minimal role in patient-reported effects. The surface heat flux increase might be more relevant to subjective patency than the magnitude in the present flow configuration, as the data in Table 7 suggest. This observation would conveniently eliminate the cumbersome task of evaluating the latent heat in assessing the physiological outcomes of surgery.

The results of this study have several limitations. Because only eight patients were considered, mapping of CFD results to patient-reported patency should be interpreted with caution. A further limitation is associated with the assumption of steady flow with a constant inspiratory airflow rate. During the respiratory cycle, the airflow rate changes continuously, but several studies [24, 25] suggest that steady-state solutions reasonably represent the overall features of the transient solution. Therefore, our present methodology is suitable for revealing the essential features of nasal airflow before and after septoturbinoplasty. Another limitation of the present study is the exclusion of the subjects affected by nasal cycle. In the absence of reference studies standardizing the effect of nasal cycle on the nasal airflow, we decided to minimize the compounding effect of nasal cycle on the airflow. Therefore, our results should be interpreted with this limitation.

This study was conducted on patients who underwent septoplasty and turbinoplasty together. Still, there is a plan to research to elucidate the proper indications for each procedure in the future.

## Conclusion

Correlations between improvements in a patient's sensation of patency and CFD-derived variables were investigated for eight subjects who underwent septoturbinoplasty. Improvements in the NOSE score correlated strongly with a reduction in unilateral NR in the preoperative MOS ($r = 0.81$) and with an increase in surface heat flux in the domain between the nasal valve and the choanae ($r = 0.87$) on the same side. This indicates that both NR and mucous cooling affect the subjective sensation of patency. Also, the results suggest that more attention should be given to the domain of the nasal cavity between the nasal valve and the choanae when improving patients' airway patency.

## Author Contributions

**Conceptualization:** Yang Na, Hyo Yeol Kim, Yong Gi Jung.

**Data curation:** Yang Na, Youn-Ji Kim, Hyo Yeol Kim, Yong Gi Jung.

**Formal analysis:** Yang Na, Youn-Ji Kim, Yong Gi Jung.

**Funding acquisition:** Yang Na.

**Investigation:** Yang Na, Youn-Ji Kim, Hyo Yeol Kim, Yong Gi Jung.

**Methodology:** Yang Na, Youn-Ji Kim, Hyo Yeol Kim, Yong Gi Jung.

**Project administration:** Yang Na, Yong Gi Jung.

**Resources:** Yang Na, Youn-Ji Kim, Yong Gi Jung.

**Software:** Yang Na, Youn-Ji Kim.

**Supervision:** Yang Na, Hyo Yeol Kim, Yong Gi Jung.

**Validation:** Yang Na, Youn-Ji Kim, Hyo Yeol Kim, Yong Gi Jung.

**Visualization:** Yang Na, Youn-Ji Kim, Hyo Yeol Kim, Yong Gi Jung.

**Writing – original draft:** Yang Na, Youn-Ji Kim, Hyo Yeol Kim, Yong Gi Jung.

**Writing – review & editing:** Yang Na, Yong Gi Jung.

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
