## [Decision Letter · Decision Letter 0]

8 Aug 2022

PONE-D-22-14744Improvements in airflow characteristics and effect on the NOSE score after septoturbinoplasty: A computational fluid dynamics analysisPLOS ONE

Dear Dr. Jung,

Thank you for submitting your manuscript to PLOS ONE. After careful consideration, we feel that it has merit but does not fully meet PLOS ONE’s publication criteria as it currently stands. Therefore, we invite you to submit a revised version of the manuscript that addresses the points raised during the review process.

We look forward to receiving your revised manuscript.

Kind regards,

Mohammad Mehdi Rashidi

Academic Editor

PLOS ONE

Journal Requirements:

Reviewers' comments:

Reviewer's Responses to Questions

**Comments to the Author**

1. Is the manuscript technically sound, and do the data support the conclusions?

Reviewer #1: Partly

Reviewer #2: Yes

2. Has the statistical analysis been performed appropriately and rigorously? 

Reviewer #1: Yes

Reviewer #2: Yes

3. Have the authors made all data underlying the findings in their manuscript fully available?

Reviewer #1: Yes

Reviewer #2: Yes

4. Is the manuscript presented in an intelligible fashion and written in standard English?

Reviewer #1: Yes

Reviewer #2: Yes

5. Review Comments to the Author

Reviewer #1: Dear authors,

Here are some of my comments

1. The introduction does not mentioned about the indication of turbinoplasty in combination of septoplasty. The introduction more focused of nasal septal deviation, but does not mentioned the condition of the turbinates.

2. In methodology, the patient selection on the surgery for turbinoplasty were not clear. The status of the turbinate was not mentioned and the indication of turbinoplasty was not stated.

3. The exclusion of patients with mucosal swelling due to nasal cycle was biased.

4. This study compare the objective outcome pre and postoperatively and correlate with the symptoms score. Since the inclusion was not clearly mention especially in the indication of doing turbinoplasty together with septoplasty, it will be better to compare 2 arms : a) the septotubinoplasty b) septoplasty only - if the main selection criteria was nasal septal deviation.

5. The results were clearly stated and well documented.

Reviewer #2: It is a well-established entity that proper surgical treatment of NSDs and/or turbinate pathologies doesn’t always provide adequate patient satisfaction of obstruction relief. Therefore, I think we need to more collaborate with the engineering discipline to elucidate and to better understand the underlying pathological mechanism between nasal anatomical pathologies and sense of obstruction. The present study is a good example of such collaboration. Although, the number of participants was low and further studies investigating the long-term results of such modelling are an issue of concern, the study is suitable for publication.

With my kind regards,

6. PLOS authors have the option to publish the peer review history of their article (what does this mean?). If published, this will include your full peer review and any attached files.

Reviewer #1: No

Reviewer #2: No

---

## [Author Response · Author response to Decision Letter 0]

24 Aug 2022

Rebuttal letter

Manuscript Number: PONE-D-22-14744

Dear Editor-in-Chief of PLOS ONE,

We are pleased to have the chance to revise our manuscript entitled “Improvements in 

airflow characteristics and effect on the NOSE score after septoturbinoplasty: A computational fluid dynamics analysis.” 

We appreciate the reviewers’ comments, which were very helpful in improving the paper. We have tried to address all the comments raised by the academic editor and two outstanding reviewers. We hope that we satisfyingly addressed those issues and that our revised manuscript will be suited for publication.

Sincerely, 

Corresponding author: Yong Gi Jung, MD, PhD

Address: Department of Otorhinolaryngology, Head and Neck Surgery, Samsung Seoul Hospital, 81 Irwon-Ro Gangnam-gu. Seoul, Korea, 06351

Tel.: (82) 02-3410-3579

Fax.: (82) 02-3410-3879

E-mail: Ent.jyg@gmail.com

Reply to Academic editor

1. Journal Style; The authors wrote the manuscript according to the journal style regulations of PLOE ONE. Additionally, although not pointed out by reviewers, several awkward phrases were corrected in the manuscript and highlighted.

2. IRB; This study was conducted after obtaining approval from the Samsung medical center IRB. All patient data used in this study were completely anonymized, and the institutional anonymization procedures were followed. Patient data were collected retrospectively, and informed consent was waived after the IRB review. Details were added to the manuscript and marked with highlights.

3. Data availability statement; All physical data acquired from CFD analysis were attached in detail to the manuscript of this study. We believe there is no additional data set to be submitted.

Reply to Reviewer #1

The authors appreciate the reviewer’s comments, which helped improve the paper. We have tried to address the reviewer’s comments to the best of our knowledge in this report and the revised manuscript. In summary, the main changes made in the revised manuscript are outlined in the following table. 

List of issues Original manuscript Revised manuscript

Septoplasty/

Turbinoplasty Insufficient information - In the introduction section, a description on septoplasty and turbinoplasty has been added 

Bias issue of the nasal cycle Insufficient information - In the discussion section, the limitation of the exclusion of the patients affected by the nasal cycle has been added

Responses to specific comments:

1. The introduction does not mention about the indication of turbinoplasty in combination of septoplasty. The introduction more focused of nasal septal deviation but does not mention the condition of the turbinates.

We agree with the reviewer's point. Septoplasty and turbinoplasty are performed together in almost all patients; therefore, they are commonly called septoturbinoplasty. To help readers better understand our results, the following explanation has been added to the “Introduction” section of the revised manuscript. We appreciate this reviewer’s helpful comments. 

“Septoplasty is a procedure that corrects an asymmetric nasal septum to the midline, and turbinoplasty is a procedure that reduces the volume of the inferior turbinate. In almost all NSD patients, septoplasty and turbinoplasty are performed together; therefore, the term septoturbinoplasty is widely used.”

2. In methodology, the patient selection on the surgery for turbinoplasty was not clear. The status of the turbinate was not mentioned and the indication of turbinoplasty was not stated.

The primary purpose of performing turbinoplasty is to prevent the concave nasal airway from narrowing after septoplasty. Therefore, turbinoplasty is frequently performed together with septoplasty, and there is no specific indication of turbinoplasty. In the present study, turbinoplasty was performed with septoplasty in all the patients (described in the “Patient recruitment and surgical techniques” section). We fully understand the reviewers' concern on this issue and hope that our explanation avoids any confusion. We believe that additional description on this issue is not necessary in the revised manuscript. 

3. The exclusion of patients with mucosal swelling due to nasal cycle was biased. 

We fully understand the reviewer’s concern regarding the issue of the nasal cycle. Judging from this comment, the reviewer seems to have a very high level of insight into these studies. The term ‘nasal cycle’ describes spontaneous changes in the engorgement of the nasal mucosa that results in fluctuations in the unilateral nasal resistance and airflow. Although 3-fold reduction in unilateral nasal resistance are often achieved by nasal surgery (ref. A1-A4), unilateral resistance may oscillate up to 5-fold during the nasal cycle (ref. A5), thus obscuring the quantification of the surgical effect. Therefore, the nasal cycle represents a significant challenge when comparing pre-and postoperative objective measures of nasal airflow via CFD analysis. At present, the effect of the nasal cycle on the airflow change has not been well-established (ref. A6). In this context, several CFD studies on the airflow modification after the surgery attempted to minimize the compounding effects of the nasal cycle on modeling results and therefore did not include the subjects affected by nasal cycle (ref. A7, A8). Ref. A9 also discussed that the nasal cycle effect is a limitation of their CFD study. 

We also discussed this issue seriously at the beginning of this study. In the absence of reference studies to standardize the effect of the nasal cycle on the nasal airflow, we decided to exclude the compounding effect resulting from the nasal cycle as in the references investigating the airflow modification resulting from nasal surgery (A7-A8). Therefore, three otolaryngologists who did not participate in this study reviewed the patients’ CT scans and determined the presence of a nasal cycle, which has been described in the “Patient recruitment and surgical techniques” section of the original manuscript. As a researcher, we did our best to minimize bias, and we sincerely hope that reviewers will understand our consideration on this issue. To reflect the reviewer’ concern, we described the limitation of our study regarding the nasal cycle in the “Discussion” session of the revised manuscript as follows:

“Another limitation of the present study is the exclusion of the subjects affected by nasal cycle. In the absence of reference studies standardizing the effect of the nasal cycle on the nasal airflow, we decided to minimize the compounding effect of the nasal cycle on the airflow. Therefore, our results should be interpreted with this limitation. 

4. This study compares the objective outcome pre and postoperatively and correlate with the symptoms score. Since the inclusion was not clearly mentioned especially in the indication of doing turbinoplasty together with septoplasty, it will be better to compare 2 arms : a) the septoturbinoplasty b) septoplasty only - if the main selection criteria were nasal septal deviation.

We again appreciate these thoughtful comments. As described in the answers to questions No. 1 and 2, turbinoplasty and septoplasty are performed together in almost all patients in current medical practice. And the authors, as clinical physicians, believe it is a right approach to combine both procedures to provide maximum post-operative patient satisfaction. Since septoturbinoplasty was performed in all the patients of this study (as described in “Patient recruitment and surgical techniques” section), it is not possible to divide the present study group into two sub-groups (one group that received turbinoplasty and the other group that did not). We hope that our explanation is a sufficient answer to the author's comment.

5. The results were clearly stated and well documented. 

We really appreciate the kind comment. 

References used in the rebuttal letter

A1. Broms P, Jonson B, Malm L. Rhinomanometry, IV: a pre- and postoperative evaluation in functional septoplasty. Acta Otolaryngol. 1982; 94:523-529.

A2. Jessen M, Ivarsson A, Malm L. Nasal airway resistance and symptoms after functional septoplasty: comparison of findings at 9 months and 9 years. Clin Otolaryngol Allied Sci. 1989; 14:231-234.

A3. Jessen M, Malm L. The importance of nasal airway resistance and nasal symptoms in the selection of patients for septoplasty. Rhinology. 1984; 22:157-164.

A4. Moore M, Eccles R. Objective evidence for the efficacy of surgical management of the deviated septum as a treatment for chronic nasal obstruction: a systematic review. Clin Otolaryngol. 2011; 36:106-113.

A5. Hasegawa M, Kern EB. Variations in nasal resistance in man: a rhinomanometric study of the nasal cycle in 50 human subjects. Rhinology. 1978; 16:19-29.

A6. Patel RG, Garcia GJM, Frank-Ito DO, Kimbell JS, Rhee JS. Simulating the nasal cycle with computational fluid dynamics. Otolaryngology-Head Neck Surg. 2015; 15(2):353-360. 

A7. Kimbell JS, Garcia GJM, Frank DO, Cannon DE, Pawar SS, Rhee JS. Computed nasal resistance compared with patient-reported symptoms un surgically treated nasal airway passages: A preliminary report. Am J Rhinol Allergy. 2012; 26(3):e94-e98.

A8. Kimbell JS, Frank DO, Laud P, Garcia GJM, Rhee JS. Changes in nasal airflow and heat transfer correlate with symptom improvement after surgery for nasal obstruction. J Biomech. 2013; 46:2634-2643.

A9. Zhao K, Jiang J. What is normal nasal airflow? A computational study of 22 healthy adults. Int Forum Allergy Rhinol. 2014; 4(6): 435-446. 

Reply to Reviewer #2

1. It is a well-established entity that proper surgical treatment of NSDs and/or turbinate pathologies doesn’t always provide adequate patient satisfaction of obstruction relief. Therefore, I think we need to more collaborate with the engineering discipline to elucidate and to better understand the underlying pathological mechanism between nasal anatomical pathologies and sense of obstruction. The present study is a good example of such collaboration. Although, the number of participants was low and further studies investigating the long-term results of such modeling are an issue of concern, the study is suitable for publication.

We appreciate this reviewer’s encouraging comment to our collation work with CFD engineers. As the reviewer understands well, CFD study on the nasal airflow requires significant amount of computational expense performing multi-step procedures (i.e., segmentation, computational modeling, high-performance computing and post-processing work). In the present study, simulations for pre- and postoperative states were conducted for each patient, resulting in a total of 16 CFD simulations for 8 patients. In addition, at least 10-20 more preliminary CFD simulations needed to be conducted to determine the pressure boundary conditions to be described at the exit. Therefore, even with 8 patients, computational cost is quite large. As the reviewer pointed out, the sample size of our study is not considered to be large, but we still believe that our study was able to reasonably capture important features of nasal physiology in terms of objective variables that are correlated to the improvement of subjective sense of patency. Nevertheless, as the reviewer suggested, we will try to increase the sample size in the future study, and we hope you will keep an eye on our future research.

---

## [Decision Letter · Decision Letter 1]

22 Sep 2022

PONE-D-22-14744R1Improvements in airflow characteristics and effect on the NOSE score after septoturbinoplasty: A computational fluid dynamics analysisPLOS ONE

Dear Dr. Jung,

Thank you for submitting your manuscript to PLOS ONE. After careful consideration, we feel that it has merit but does not fully meet PLOS ONE’s publication criteria as it currently stands. Therefore, we invite you to submit a revised version of the manuscript that addresses the points raised during the review process.

We look forward to receiving your revised manuscript.

Kind regards,

Mohammad Mehdi Rashidi

Academic Editor

PLOS ONE

Reviewers' comments:

Reviewer's Responses to Questions

**Comments to the Author**

1. If the authors have adequately addressed your comments raised in a previous round of review and you feel that this manuscript is now acceptable for publication, you may indicate that here to bypass the “Comments to the Author” section, enter your conflict of interest statement in the “Confidential to Editor” section, and submit your "Accept" recommendation.

Reviewer #1: (No Response)

Reviewer #2: All comments have been addressed

2. Is the manuscript technically sound, and do the data support the conclusions?

Reviewer #1: Yes

Reviewer #2: Yes

3. Has the statistical analysis been performed appropriately and rigorously? 

Reviewer #1: Yes

Reviewer #2: Yes

4. Have the authors made all data underlying the findings in their manuscript fully available?

Reviewer #1: (No Response)

Reviewer #2: Yes

5. Is the manuscript presented in an intelligible fashion and written in standard English?

Reviewer #1: Yes

Reviewer #2: Yes

6. Review Comments to the Author

Reviewer #1: The indication of septoplasty and turbinoplasty were not clearly stated in all patients. Does all patients need both procedures? A control arm of septoplasty alone and turbinoplasty alone should be done to compare the improvement of combined procedures.

Reviewer #2: I have no competing interests. All comments have been addressed and the paper is suitable for publication (accepted).

7. PLOS authors have the option to publish the peer review history of their article (what does this mean?). If published, this will include your full peer review and any attached files.

Reviewer #1: **Yes: **Farah Dayana Zahedi

Reviewer #2: No

---

## [Author Response · Author response to Decision Letter 1]

24 Sep 2022

Rebuttal letter

Manuscript Number: PONE-D-22-14744R1

Dear Editor-in-Chief of PLOS ONE,

We have revised our manuscript entitled “Improvements in airflow characteristics and effect on the NOSE score after septoturbinoplasty: A computational fluid dynamics analysis.” 

In the revised paper, we have tried to address the comments raised by the reviewer and 

the editor. We found those comments helpful once again. We hope that we satisfyingly addressed those issues and that our revised manuscript will be suited for publication.

Sincerely, 

Corresponding author: Yong Gi Jung, MD, PhD

Address: Department of Otorhinolaryngology, Head and Neck Surgery, Samsung Seoul Hospital, 81 Irwon-Ro Gangnam-gu. Seoul, Korea, 06351

Tel.: (82) 02-3410-3579

Fax.: (82) 02-3410-3879

E-mail: Ent.jyg@gmail.com

Reply to the Academic editor

1. Have the authors made all data underlying the findings in their manuscript fully available?

The PLOS Data policy requires authors to make all data underlying the findings described in their manuscript fully available without restriction, with rare exception (please refer to the Data Availability Statement in the manuscript PDF file). The data should be provided as part of the manuscript or its supporting information or deposited to a public repository. For example, in addition to summary statistics, the data points behind means, medians and variance measures should be available. If there are restrictions on publicly sharing data—e.g., participant privacy or use of data from a third party—those must be specified.

(Response)

To support author compliance of the PLOS data policy, we have deposited our data to the public repository of Dryad (which is recommended by PLOS One). Our data can be accessed by the following link: https://orcid.org/0000-0001-5729-6912. In this repository, we have deposited the following 4 files (one readme file and three datafile). The readme file (Microsoft Word format) summarizes the title, authors, abstract and the data files. Three datafiles (Microsoft Excel format) contain the raw data used in the present study. 

(1) README File

(2) flow_partition_nasal_resistance_data_for_Dryad

- Flow partition, unilateral nasal resistance, bilateral nasal resistance of the pre- and postoperative models

(3) relative_humiduty_air_temperature_data_for_Dryad

- Relative humidity, air temperature of the pre- and postoperative models

(4) surface_heat_flux_data_for_Drayad

- Surface heat flux (sensible and latent heat) in the MOS and LOS of the Area-1 of the pre- and postoperative models

- Surface heat flux (sensible and latent heat) in the MOS and LOS of the Area-2 of the pre- and postoperative models

We have indicated the above data deposition in the first page of the revised paper as follows:

“Data availability: Data for this study are available in DRYAD at https://doi.org/10.5061/dryad.nk98sf7wj “

Reply to Reviewer #1

The authors appreciate this reviewer’s comment once again. We have tried to address the reviewer’s comment more clearly to the best of our knowledge in this report and the revised manuscript. 

Responses to specific comments:

1. Reviewer #1: The indication of septoplasty and turbinoplasty were not clearly stated in all patients. Do all patients need both procedures? A control arm of septoplasty alone and turbinoplasty alone should be done to compare the improvement of combined procedures.

(Response)

Thank you very much for your thoughtful comments.

The following sentence has been added to the methods section so that readers can more clearly understand the patient group selection in this study.

 “Both septoplasty and turbinoplasty were recommended to secure the patient's nasal airway as much as possible, and septoturbinoplasty was performed in all subjects after obtaining consent.”

And the research ideas suggested by the reviewers were added to the discussion section as in the following sentence and marked with highlighted.

“This study was conducted on patients who underwent septoplasty and turbinoplasty together. Still, there is a plan to research to elucidate the proper indications for each procedure in the future.”

The purpose of this study was not to compare the pattern of CFD change before and after surgery in septoplasty and turbinoplasty, but to compare symptom improvement and CFD parameters of patients who underwent both procedures. Therefore, please understand the purpose of this study.

However, the reviewer's opinions are clinically critical, and I think it would be meaningful to separate and compare septoplasty and turbinoplasty. Because septoturbinoplasty is performed in almost all patients at our institution, it is difficult to conduct a retrospective study on the above topic, but a prospective study can be possible.

 Thank you for giving me a good idea, and I will plan further research on the above topic. So please stay tuned for further study by our team. 

Appreciate your helpful comment.

---

## [Editor Report · Decision Letter 2]

2 Nov 2022

Improvements in airflow characteristics and effect on the NOSE score after septoturbinoplasty: A computational fluid dynamics analysis

PONE-D-22-14744R2

Dear Dr. Jung,

We’re pleased to inform you that your manuscript has been judged scientifically suitable for publication and will be formally accepted for publication once it meets all outstanding technical requirements.

Kind regards,

Mohammad Mehdi Rashidi

Academic Editor

PLOS ONE
---

## [Editor Report · Acceptance letter]

8 Nov 2022

PONE-D-22-14744R2 

Improvements in airflow characteristics and effect on the NOSE score after septoturbinoplasty: A computational fluid dynamics analysis 

Dear Dr. Jung:

I'm pleased to inform you that your manuscript has been deemed suitable for publication in PLOS ONE. Congratulations! Your manuscript is now with our production department. 

Kind regards, 

on behalf of

Professor Mohammad Mehdi Rashidi 

Academic Editor

PLOS ONE